# Stylebreeder 🎨: Exploring and Democratizing Artistic Styles through Text-to-Image Models

**Matthew Zheng**[1,*]   **Enis Simsar**[2,*]   **Hidir Yesiltepe**[1]

**Federico Tombari**[3,4]   **Joel Simon**[5]   **Pinar Yanardag**[1]

[1]Virginia Tech       [2]ETH Zürich       [3]TUM       [4]Google       [5]Artbreeder

https://stylebreeder.github.io

## Abstract

Text-to-image models are becoming increasingly popular, revolutionizing the landscape of digital art creation by enabling highly detailed and creative visual content generation. These models have been widely employed across various domains, particularly in art generation, where they facilitate a broad spectrum of creative expression and democratize access to artistic creation. In this paper, we introduce STYLEBREEDER, a comprehensive dataset of 6.8M images and 1.8M prompts generated by 95K users on Artbreeder, a platform that has emerged as a significant hub for creative exploration with over 13M users. We introduce a series of tasks with this dataset aimed at identifying diverse artistic styles, generating personalized content, and recommending styles based on user interests. By documenting unique, user-generated styles that transcend conventional categories like 'cyberpunk' or 'Picasso,' we explore the potential for unique, crowd-sourced styles that could provide deep insights into the collective creative psyche of users worldwide. We also evaluate different personalization methods to enhance artistic expression and introduce a style atlas, making these models available in LoRA format for public use. Our research demonstrates the potential of text-to-image diffusion models to uncover and promote unique artistic expressions, further democratizing AI in art and fostering a more diverse and inclusive artistic community. The dataset, code, and models are available at https://stylebreeder.github.io under a Public Domain (CC0) license.

## 1   Introduction

Text-to-image models, such as Denoising Diffusion Models (DDMs) [19] and Latent Diffusion Models (LDMs) [39], are becoming increasingly popular and are revolutionizing the landscape of digital art creation. These models are renowned for their capability to generate high-quality, high-resolution images across diverse domains, enabling the production of highly detailed and creative visual content [59, 8, 21, 60, 12, 32, 25]. Artists worldwide are increasingly leveraging these models, utilizing diverse textual prompts to create artworks spanning myriad styles, thereby democratizing the art creation process and making it more accessible.

In the midst of this technological renaissance, platforms like Artbreeder have surfaced as pivotal hubs for creative exploration. Artbreeder, with its user base exceeding 13 million, facilitates the generation of millions of images that embody a vast spectrum of artistic styles.

---

[*]Joint first authors.

38th Conference on Neural Information Processing Systems (NeurIPS 2024) Track on Datasets and Benchmarks.

| **Image** | **Prompts** | **Additional Features** | | | |
|---|---|---|---|---|---|

**Positive Prompt**

black and red and gold shot of a meadow in spring, flowers everywhere, lush green grass, beautiful sky, realistic faces, girl in a flowy dress and flowers in her arms, dark fantasy, intricate details, hyper detailed, jean baptiste monge, carne griffiths, michael garmash, tsutomu nihei, motifs of spring and renewal : 1

**Negative Prompt**

bad hands, amateur, low quality, mangled hands, disfigured hands, ugly hands : -0.5

| | | |
|---|---|---|
| **ImageID** 298709787 | **UserID** 174769 | |
| **Model** sdxl-1.0 | **Step** 13 | |
| **Image Size** (640, 960) | **Seed** 556 | |
| **Timestamp** 2024-03-20 3:02:23 | **CFG Scale** 26 | |

| **Identity Attack** 0.00456 | **Prompt NSFW** 0.00015 |
|---|---|
| **Insult** 0.0031 | **Image NSFW** 0.00573 |
| **Threat** 0.001 | **Toxicity** 0.00364 |
| **Cluster ID** 3398 | **Obscene** 0.00037 |

Figure 1: Our dataset comprises 6.8M images generated by 95,000 unique users, accompanied by 1.8M text prompts from July 2022 to May 2024. It includes detailed metadata such as Positive Prompt, Negative Prompt, UserID, Timestamp, and Image Size. Additionally, we supply model-related hyperparameters, including Model Type, Seed, Step, and CFG Scale. Note that the disparity in prompts and images arises because different images can be generated from the same text prompt when varying hyperparameters. We also offer further metadata like Cluster ID, along with scores for Prompt NSFW, Image NSFW, and Toxicity computed using state-of-the-art models [15, 16].

This surge in user-generated content presents an intriguing question: beyond the conventional styles typically prompted by terms like 'cyberpunk' or 'Picasso,' what unique, crowd-sourced styles might exist within such a community? These styles, potentially undocumented and uniquely communal, could offer profound insights into the collective creative psyche of users worldwide.

However, existing datasets often fall short in terms of exploring the visual potential of user-generated images. While some works similar to ours, such as Diffusion DB [57] and TWIGMA [5] also explore AI-generated content, they either feature a smaller user base, do not provide original text prompts associated with the images, or provide limited insights into artistic styles from a visual perspective. Addressing this inquiry, our paper introduces a comprehensive dataset derived from Artbreeder, providing contributions from 95K unique users, 6.8M images and 1.8M text prompts generated with a variety of text-to-image diffusion models such as Stable Diffusion (SD) [39] or Stable Diffusion XL (SD-XL) [36] (see Fig. 1). We identified and analyzed novel, user-generated artistic styles, uncovering diverse and previously unrecognized creative expressions. Based on the discovered styles, we showcase a variety of tasks, including personalization and recommendation. Our contributions are as follows:

- We present an extensive dataset, STYLEBREEDER, from Artbreeder on CC0 license, capturing millions of user-generated images and styles and sharing them with the community to further encourage research in this area.

- We cluster images based on their stylistic similarities, helping to map out the landscape of user-generated art.

- Utilizing historical data of the users, we showcase a recommendation system that aligns style suggestions with individual preferences, making the exploration of artistic styles more targeted and meaningful.

- We release a web-based platform, Style Atlas, providing public access to download pre-trained style LoRAs for personalized content generation, encouraging experimentation and collaborative artistic exploration to expand our understanding of digital creativity and further democratize the creative use of AI.

## 2 Related Work

We provide a brief overview of artwork datasets, followed by discussions on text-to-image diffusion models and personalized image generation.

Table 1: A comparison of our dataset to other AI-generated image datasets

| Name | Source | # Images | Year | Original Prompt Included | Multiple Models |
|------|--------|----------|------|--------------------------|-----------------|
| DiffusionDB | SD Discord | 14,000,000 | Aug 2022 | ✓ | ✗ |
| Midjourney Kaggle | Midjourney | 250,000 | Jun 2022-Jul 2022 | ✓ | ✗ |
| TWIGMA | Twitter | 800,000 | Jan 2020-Mar 2023 | ✗ | ✓ |
| **STYLEBREEDER (Ours)** | Artbreeder | 6,818,217 | Jul 2022-May 2024 | ✓ | ✓ |

## 2.1 Artwork Datasets

Traditional artwork datasets (see Supplemental Materials for a comparison) have primarily focused on artwork classification and attribute prediction. However, these datasets often exhibit limitations, like skewed class distributions and unsuitable classes for image synthesis, when employed for artwork synthesis. To address these shortcomings in evaluating artwork synthesis, specialized subsets have been curated to better suit the task. For instance, [52] and [61] derived datasets by scraping WikiArt images. Despite these efforts, such datasets still face many challenges related to variable image quality, imbalanced distributions, and others. ArtBench-10 [28] attempted to rectify these issues by introducing a class-balanced and cleanly annotated benchmark. However, it only provides ten classes.

Recent advancements in text-to-image synthesis have spurred the development of AI-generated datasets like DiffusionDB [57], and Midjourney Kaggle [33], which contain millions of image-text pairs generated by models such as Stable Diffusion and Midjourney. These datasets, while groundbreaking, tend to be limited in stylistic diversity and are skewed towards specific user groups, reflecting data collected from constrained environments and short time frames. The main distinction between our dataset and those datasets lies in the duration over which the images, along with the magnitude of the images. While the DiffusionDB dataset covers a brief period of just 12 days in August 2022, our dataset extends across a much longer time frame, spanning 18 months from July 2022 to May 2024. This extensive duration provides a significant advantage for in-depth studies into the evolution and dynamics of visual trends, artistic styles, and thematic content. Researchers have tailored other datasets to investigate certain themes [4, 27, 47, 56, 49, 31, 58], but these are domain-specific and lack breadth. TWIGMA [5] captured multiple years of generated images scraped from X (formerly Twitter) but does not include the prompts that were used to generate these images, instead relying on inferred BLIP captions. As shown in Tab. 1, our dataset not only includes the original prompts used to produce images but also contains images generated by multiple models while encapsulating a long time frame.

## 2.2 Text-to-image Generative Models

Text-to-image generative models (T2I) [42, 37, 39] have displayed exceptional abilities in synthesizing high-quality images conditioned on textual descriptions. As they improve, these models become increasingly indispensable tools for visually creative tasks. Among these, diffusion-based text-to-image models [20, 2, 48] are prominently used for guided image synthesis [8, 21, 60] and complex image editing [59, 3] applications. Although these models allow for significant control through text, such as directing the color or attributes of styles within generated images, they are unable to preserve the same precise style across new contexts consistently.

## 2.3 Personalized Generation in Diffusion Models

Personalization techniques have been proposed to enable pre-trained text-to-image models to generate novel concepts based on a small set of images. DreamBooth [40] fine-tunes the full T2I model, which yields more detailed and expressive outputs. However, due to the large scale of these models, full fine-tuning is an expensive task that requires substantial amounts of memory. Different methods attempt to work around this challenge for both style and content representations. Custom Diffusion [26] attempts multi-concept learning but requires expensive joint training and struggles with style disentanglement. Textual Inversion [11] learns a new token embedding to represent a subject or style without altering the original

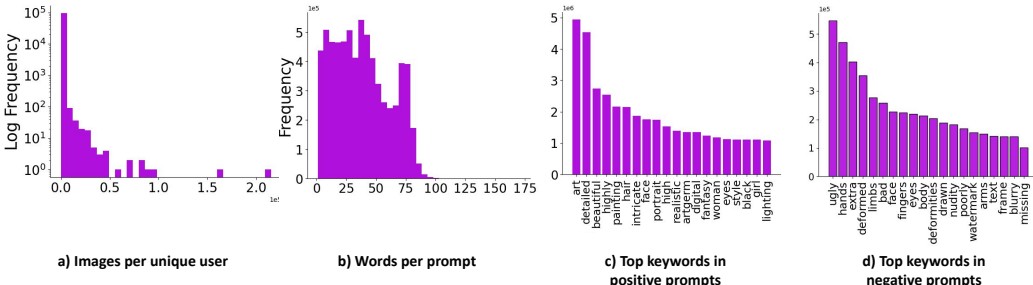

| a) Images per unique user | b) Words per prompt | c) Top keywords in positive prompts | d) Top keywords in negative prompts |

Figure 2: Most unique users have fewer than 1000 images generated. The average number of words in a prompt is less than 60 words. Common keywords for positive prompts include 'painting', 'realistic', and 'digital' reveal semantic information about the style of desired images. Common keywords in negative prompts, such as 'ugly' and 'deformed,' indicate undesired features of generated images.

model parameters, while StyleDrop [45] utilizes adapter tuning to only train a subset of weights for style adaptation. Low-Rank Adaptation (LoRA) [22] is a Parameter Efficient Fine-Tuning (PEFT) technique frequently used to fine-tune T2I models to generate images of a desired style. EDLoRA [14] proposes a layerwise embedding and multi-word representation when training a LoRA model.

## 3 Stylebreeder Dataset

We collect STYLEBREEDER by scraping images from the Artbreeder website. We choose Artbreeder since it is one of the most popular platforms for art generation, supporting various text-to-image models. Artbreeder enables users to create images using text prompts, offering controls over various settings, such as the strength (guidance scale) of the text's influence on the generated image, seed values, model type, and other hyperparameters. Since its rise in popularity within the artistic community in 2018, Artbreeder has become known for its bias towards generating artistic images. This predisposition towards artistic styles is a primary reason we concentrated our focus on this area. Additionally, all images on Artbreeder are covered by a CC0 license[2], which allows for unrestricted use for any purpose [3]. We collect metadata along with the images, which include text prompts (positive and negative), usernames, and hyperparameters. We provide additional features such as NSFW scores for each image and text prompts (see Fig. 1).

### 3.1 User Statistics

Our dataset comprises 95,479 unique users, each generating an average of 72.41 images. Figure 2 (a) illustrates the distribution of images per user, showing that the majority of users produced fewer than 1,000 images, although a few power users created a significantly larger number of images. All user IDs in our dataset have been anonymized to ensure users' privacy.

### 3.2 Model Statistics

Our dataset represents a wide variety of text-to-image diffusion models, including Stable Diffusion 1.5 (74.9%), SD-XL 1.0 (13.1%), Stable Diffusion 1.3 (8.8%), Stable Diffusion 1.4 (1.3%), Stable Diffusion 1.5-free (1.1%) and ControlNet 1.5 (0.8%). These models differ in their capabilities and the quality of their generated outputs, with recent models often supporting higher resolutions that deliver finer details and more complex visuals. This variety in models used to generate images provides a valuable opportunity to explore their differences,

[2]https://creativecommons.org/public-domain/cc0
[3]https://www.artbreeder.com/terms.pdf

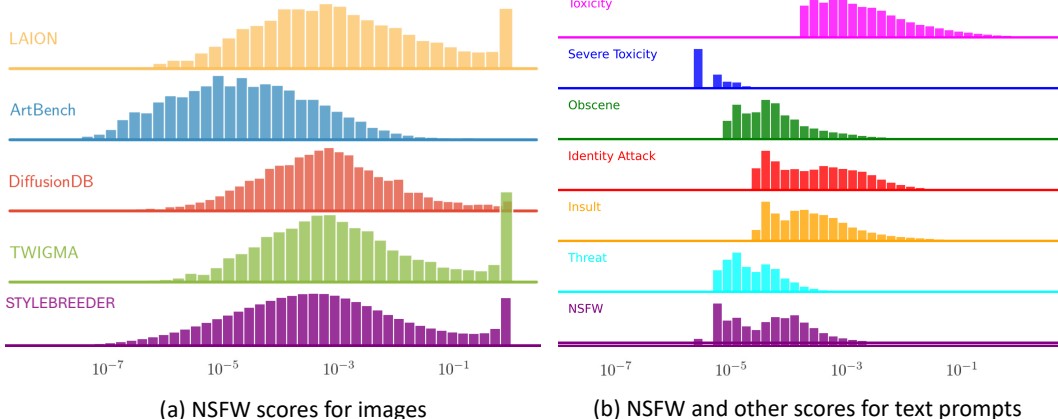

(a) NSFW scores for images  (b) NSFW and other scores for text prompts

Figure 3: (a) Predicted NSFW scores across LAION [44], Artbench [28], DiffusionDB [57] and TWIGMA [5], STYLEBREEDER (Ours) on images[4], computed with [16] (higher score indicates more NSFW content). (b) Predicted NSFW, Toxicity, Severe Toxicity, Identity Attack, Insult, and Threat scores across on text prompts, computed with [15] on STYLEBREEDER.

such as variations in artistic expression, the nuances in image quality, and potential biases inherent in each model. Stable Diffusion 1.5 is the most frequently used model in our dataset, followed by SD-XL, which has gained popularity due to its ability to generate high-quality images. The resolution of the images ranges from $512 \times 512$ to $1280 \times 896$ based on the model employed. Additionally, our dataset provides details on key hyperparameters like seed, CFG guidance scale—which dictates the extent to which the image generation process adheres to the text prompt—and step size in the diffusion model. Users are able to generate varying images using identical text prompts by adjusting these parameters. How different configurations affect the resulting images introduces deeper insights into the model's behavior and its sensitivity to these parameters. This enables a deeper understanding of how subtle changes in input or settings can significantly alter the characteristics of generated images, providing valuable perspectives on the underlying generative processes.

## 3.3 Text Prompts

We collect both positive and negative text prompts for each image in our dataset. The average prompt length is 60 words, as shown in Fig. 2 (b). Common words in positive prompts, such as 'painting', 'realistic', and 'digital', reveal semantic information about the desired styles of images (see Fig. 2 (c)) while common words in negative prompts like 'ugly' and 'deformed' indicate the undesired features of the generated images (see Fig. 2 (d)). Furthermore, we observe that users often incorporate artist names in their text prompts to specify desired styles, a common practice employed by the generative art community. To quantify this trend, we analyze using BERT NER [53] to identify unique artist names in the dataset. Our findings highlight a significant occurrence of artist names, with top mentions including 'Ilya Kuvshinov', a Russian illustrator, featured in 208K text prompts, and 'Akihiko Yoshida', a Japanese video game artist, appearing in 81K prompts. Given that these artists may not permit the use of their artistic styles, we offer a form on our website allowing artists to opt-out, ensuring their styles are not replicated without their consent, as outlined in Section 5.

## 3.4 NSFW and Toxic Content

We analyze NSFW content in both images and text prompts using state-of-the-art predictors for images [16] and text prompts [15]. Figure 3 (a) shows a comparison of our dataset (STYLEBREEDER) with other popular datasets such as LAION [44], Artbench [28], DiffusionDB

---

[4]NSFW plots for LAION, Artbench, TWIGMA, and DiffusionDB are adopted from [5].

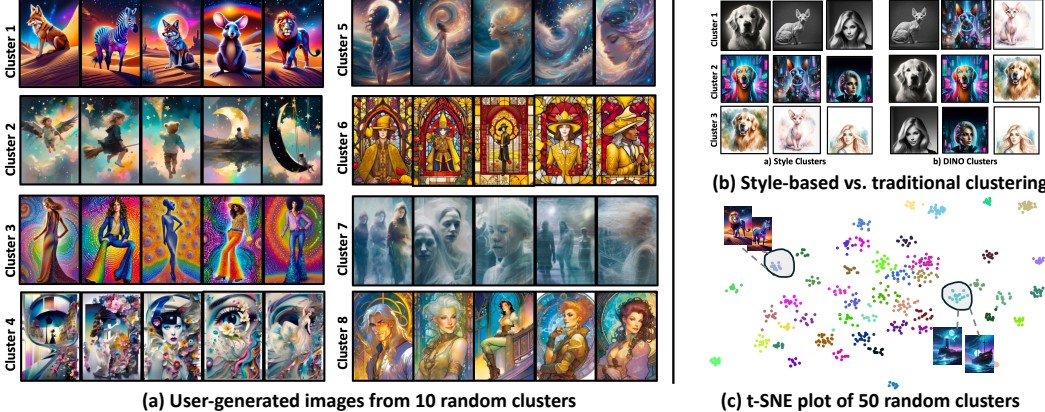

**(b) Style-based vs. traditional clustering**

**(a) User-generated images from 10 random clusters**

**(c) t-SNE plot of 50 random clusters**

Figure 4: (a) User-generated images from 10 random clusters showcasing a diverse range of styles. (b) Sample images from style-based clustering vs. traditional clustering using DINO features show that style-based clustering captures the stylistic content while traditional clustering focuses on objects. (c) Visualization of the clusters, projected into 2D with t-SNE [18] with each cluster represented by a unique color according to their assignments by K-Means++ [1]. This depiction highlights that while many styles are closely related, some distinct styles are noticeably distant from the main clusters.

[57] and TWIGMA [5]. Most of these datasets, particularly those with AI-generated images like DiffusionDB, TWIGMA, and ours, contain a substantial amount of potentially NSFW content. A similar observation can be made for text-prompts (see Fig. 3 (b)) where we report NSFW, Toxicity, Severe Toxicity, Identity Attack, Obscene, Insult, and Threat scores computed with [15]. This trend correlates with recent studies highlighting a significant increase in NSFW content generation by online communities [38, 43]. For instance, potentially harmful text prompts may involve the names of influential politicians, such as 'Donald Trump' and 'Joe Biden', found in prompts like 'angry Joe Biden screaming, red-faced, steam coming from ears' or 'angry Donald Trump Melania and the judge and police at mcdonalds'. Additionally, our analysis reveals the use of celebrity names in contexts suggesting sexually explicit content that could be construed as nonconsensual pornography. To assist researchers, our dataset includes NSFW text and image scores, along with toxicity-related scores, enabling them to filter these images and determine appropriate thresholds for excluding potentially unsafe data. We also provide a Google form for reporting harmful or inappropriate images and prompts, as outlined in Section 5.

## 4 Experiments

We define three tasks using our dataset: identifying diverse artistic styles through clustering based on stylistic similarities, generating personalized images tailored to individual styles, and recommending styles based on previously generated images of a given user.

### 4.1 Experimental Setup

We utilize official and Diffusers [35] implementations for Textual Inversion [11], LoRA w/DreamBooth [22, 40], Custom Diffusion [26], and EDLoRA [14]. For each method, we use the same set of seeds and adhered to the parameter settings recommended in their respective publications. Textual Inversion was trained at a resolution of $512 \times 512$ for 3000 steps with a learning rate of 5e-4. LoRA with DreamBooth and EDLoRA were both trained at the same resolution ($512 \times 512$) for 800 steps but with a learning rate of 1e-4, and LoRA had a rank of 32. Custom Diffusion was trained at a resolution of $512 \times 512$ for 250 steps with a learning rate of 1e-5. DINO and CLIP scores in Tab. 2 were computed on 20K images each, and standard deviations are provided. We use 8 NVIDIA L40 GPUs for our experiments. For recommendation tasks, we use 96K images generated by 1434 users with an 80% training

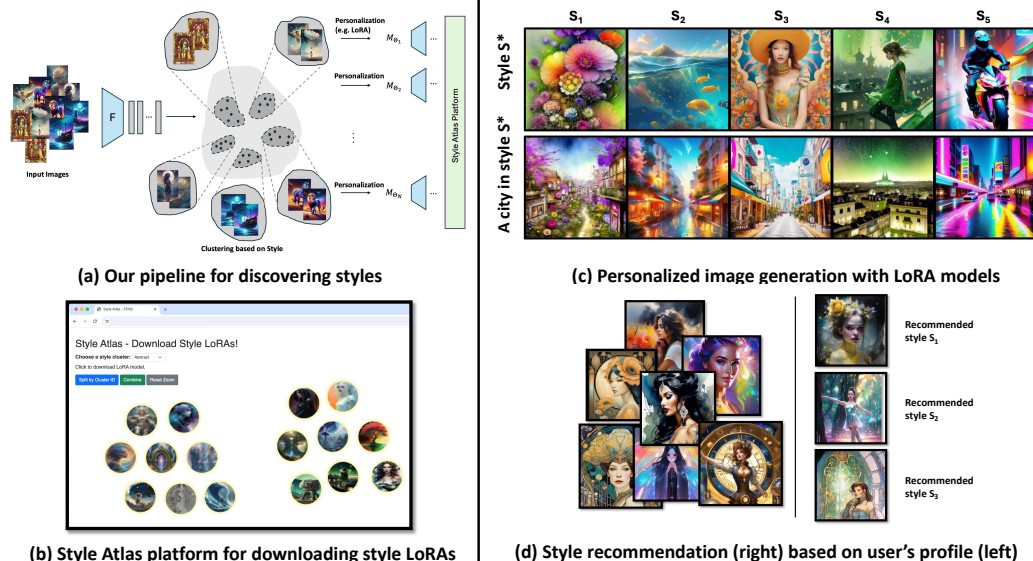

**(a) Our pipeline for discovering styles**

**(c) Personalized image generation with LoRA models**

**(b) Style Atlas platform for downloading style LoRAs**

**(d) Style recommendation (right) based on user's profile (left)**

Figure 5: (a) An illustration of our pipeline: we cluster input images by stylistic similarity and employ a personalization method, such as LoRA, to train personalized models aligned with specific styles. (b) Users can download style LoRA models from the Style Atlas platform. (c) Users can generate personalized images using LoRA models where `Style S*` represents an example image from the cluster. (d) We recommend top styles to users based on the images they have previously generated. This personalized approach helps tailor style suggestions to each user's unique preferences.

and 20% test split, using a learning rate of 5e-3 and regularization term 2e-2 and analyzing a 5-fold cross-validation using the Surprise [50] library.

## 4.2 Discovering Diverse Artistic Styles

The rising popularity of generated art has led to a vast array of user-generated content showcasing a diverse spectrum of artistic styles. However, a key challenge remains: how can we effectively discover and categorize these styles? We aim to cluster user-generated images into stylistically similar groups to uncover unique styles. Formally, our dataset, denoted as $\mathcal{D}$, consists of $N$ images. We employ a pre-trained text-to-image model $M_\Theta$, parameterized by weights $\theta$ and a token embedding space $V$. Firstly, we convert the images into a set of $N$ style embeddings using a state-of-the-art feature extractor $F$, specifically CSD [46], which uses a Vision Transformer (ViT) [9] backbone to map images into a $d$-dimensional vector space representing their style descriptors. CSD has shown superior performance in style-matching tasks across datasets like DomainNet, WikiArt, and LAION-Styles, outperforming models like DINO, which focus more on semantic content. This conversion results in a set of embeddings $Z = \cup_N F(\mathbf{x}_i)$, where each image, $\mathbf{x}_i$, is embedded in a high-dimensional semantic embedding space. These embeddings are then clustered into $k = 10000$ groups using the K-Means++ [1] algorithm, which utilizes cosine similarity to ensure cluster cohesion. To determine the optimal number of clusters, we employ the silhouette score method across various cluster sizes: 50, 100, 500, 1000, 2000, 5000, 10000, 20000 (see Supplemental Materials for more details). This experiment underscores that a configuration of 10000 clusters maximizes the silhouette score, indicating optimal internal similarity and external dissimilarity among the clusters. Figure 4 (c) presents a 2D projection of the CSD embedding space using t-SNE [18]. The visualization reveals that some clusters are closely grouped, reflecting stylistic similarities, while others are distinctly isolated, demonstrating significant stylistic variations.

Table 2: Benchmark results for state-of-the-art personalized image generator models.

| | | Textual Inversion | LoRA w/DreamBooth | EDLoRA | Custom-Diffusion |
|---|---|---|---|---|---|
| **CLIP-I** | Avg. | $0.6869 \pm 0.10$ | $0.6299 \pm 0.11$ | $\mathbf{0.6957 \pm 0.11}$ | $0.5917 \pm 0.12$ |
| | Min. | $0.6166 \pm 0.10$ | $0.5654 \pm 0.11$ | $\mathbf{0.6214 \pm 0.11}$ | $0.5324 \pm 0.11$ |
| | Max. | $0.7428 \pm 0.10$ | $0.6831 \pm 0.11$ | $\mathbf{0.7521 \pm 0.12}$ | $0.6440 \pm 0.12$ |
| **CLIP-T** | Avg. | $0.1857 \pm 0.02$ | $\mathbf{0.1896 \pm 0.02}$ | $0.1822 \pm 0.01$ | $0.1809 \pm 0.02$ |
| | Min. | $0.1555 \pm 0.02$ | $\mathbf{0.1573 \pm 0.02}$ | $0.1527 \pm 0.01$ | $0.1486 \pm 0.02$ |
| | Max. | $0.2392 \pm 0.03$ | $\mathbf{0.2663 \pm 0.03}$ | $0.2389 \pm 0.03$ | $0.2585 \pm 0.03$ |
| **DINO** | Avg. | $0.3801 \pm 0.15$ | $0.2668 \pm 0.17$ | $\mathbf{0.4125 \pm 0.18}$ | $0.2546 \pm 0.17$ |
| | Min. | $0.2581 \pm 0.13$ | $0.1682 \pm 0.14$ | $\mathbf{0.2790 \pm 0.15}$ | $0.1634 \pm 0.14$ |
| | Max. | $0.4838 \pm 0.17$ | $0.3585 \pm 0.19$ | $\mathbf{0.5246 \pm 0.19}$ | $0.3402 \pm 0.19$ |

### 4.3 Personalized Image Generation Based on Style

Personalized image generation based on style is a crucial task since it allows users to create unique, customized visuals that resonate with their specific aesthetic preferences. We picked four state-of-the-art personalization methods, namely, Textual Inversion [11], LoRA w/DreamBooth [22, 40], Custom Diffusion [26], and EDLoRA [14], to provide a benchmark analysis on the discovered styles. For this task, we randomly select 40 clusters and generated 50 images for ten text prompts, totaling 500 images per cluster for each method. To provide a benchmark on these models, we calculate several metrics: CLIP-T (image-text similarity between generated images and text prompts), CLIP-I (image-image similarity between clusters and generated images), and DINO (image-image similarity between clusters and generated images). Details of these metrics can be found in Tab. 2, which demonstrates that EDLoRA exhibits superior performance in personalized image generation due to its embedding-based approach. We also provide qualitative examples in Fig. 6.

### 4.4 Style-based Recommendation

The sheer volume of styles generated by users presents a significant challenge in navigating the landscape of artistic options available. To address this, we showcase a recommendation system that suggests top styles to users based on their previously generated images. This personalized approach is crucial as it helps users discover styles that align with their tastes and past preferences, simplifying their search among a vast array of choices. We formulate our task as a matrix-factorization-based recommendation, which involves a set of items where users rate items they have interacted with, thus creating a matrix of user-item ratings. In the context of our problem, users are the creators who generate images, and items are the clusters in which generated images are assigned. We calculate for each user $u$ a vector $\mathbf{v}_u = [r_1 \quad r_2 \quad ... \quad r_N]$ where $r_i$ represents the proportion of images that the user has generated within a specific cluster $i$ such that $\sum_{i=1}^{N} r_i = 1$. These vectors create a matrix $R$ where entries $r_{ui}$ denote the rating for user $u$ for cluster $i$. We employ the SVD algorithm [51, 10] to obtain a prediction for all $\hat{r}_{ui} = \mu + b_u + b_i + q_i^T p_u$ where $\mu$ is the global average rating. $b_u$ and $b_i$ are the user and item bias terms, respectively. $q_i$ and $p_u$ are the latent factor vectors for item $i$ and user $u$, respectively. We minimize the regularized squared error loss and update parameters using Stochastic Gradient Descent [13]. We assess the MAE and RMSE of the predicted ratings against the ground-truth ratings, obtaining a mean RMSE of 0.1425 and a standard deviation of 0.0017, along with a mean MAE of 0.082 and a standard deviation of 0.001 across all folds. Fig. 5 (d) depicts an example of recommendations for a user based on previously generated styles.

### 4.5 Style Atlas for Democratizing Artistic Styles

Since LoRA has become a popular tool for lightweight concept tuning within the community [41], we provide 100 style LoRAs in our Style Atlas platform. Fig. 5 (c) displays a screenshot of our platform where users can browse and download LoRA models for appealing styles. Additionally, we provide a notebook that enables users to load these downloaded LoRA models

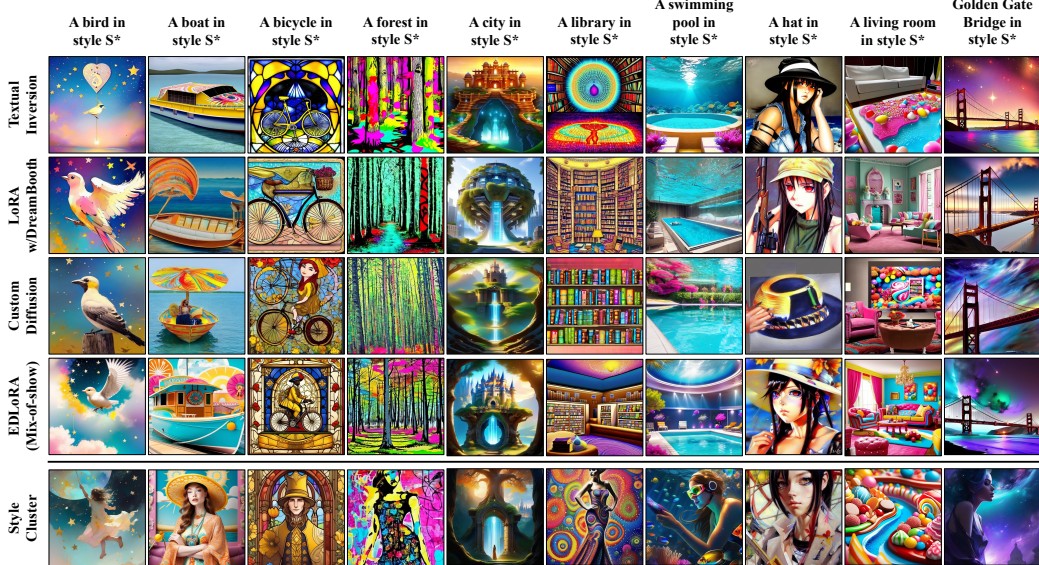

Figure 6: Qualitative comparison of various personalization methods on artistic styles on Textual Inversion [11], LoRA w/DreamBooth [22, 40], Custom Diffusion [26], and EDLoRA [14]. `Style Cluster` (bottom row) illustrates a sample image from the corresponding cluster.

and generate personalized images. For further details, please refer to the Supplementary Material.

## 5   Limitations and Societal Impact

While our work significantly advances the integration of AI in creative processes, it also presents certain limitations and societal impacts that warrant careful consideration. One of the limitations lies in the potential for over-reliance on technology in artistic creation, which could diminish the value and perception of human-driven artistry and creativity. Additionally, the use of AI in art generation raises concerns about copyright and originality, especially when styles closely mimic those of existing artists without clear attribution. From a societal perspective, while our tools aim to democratize art creation, there exists a risk of reinforcing existing biases present in the training data, which could skew the diversity and representation of generated artworks. Moreover, as these technologies become more accessible, there is a potential for misuse, such as creating deceptive images or deepfakes, which could have broader implications for trust and authenticity in digital media. Acknowledging these challenges is crucial as we continue to explore the intersection of AI and art.

## 6   Conclusion

This paper has demonstrated the significant potential of text-to-image diffusion models to explore and catalog the rich tapestry of user-generated artistic styles on the Artbreeder platform. We successfully identify unique, previously uncharted artistic expressions and demonstrate their application in generating personalized images. Additionally, we demonstrate that a personalized recommendation system enhances user engagement by aligning suggested styles with individual preferences. We also release the Style Atlas platform that democratizes access to these innovations, allowing users to experiment with and adopt new artistic expressions. This work not only advances the technological capabilities of AI in the field of art but also contributes to a more inclusive and diverse artistic community. Our dataset offers numerous avenues for further exploration, such as refining the effectiveness of text prompts through iterative adjustments, studying trends in art over time, recommending styles based on both image and textual content, developing a search system for the generated images, and exploring explainable creativity [30].

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

## A  Author Statement

We, the authors, bear all responsibility in case of violation of rights, etc., and confirm that STYLEBREEDER is available under a Public Domain (CC0) license.

## B  Dataset Hosting

Please visit our Hugging Face page for viewing and downloading the dataset. Additional information regarding STYLEBREEDER can be found at https://stylebreeder.github.io. We confirm that as authors of STYLEBREEDER, we will ensure necessary maintenance related to the dataset.

## C  Croissant Metadata

The croissant metadata can be found at https://huggingface.co/api/datasets/style-breeder/stylebreeder/croissant

## D  Style Atlas Platform

Our Style Atlas of 100 LoRAs open for download is available at https://style-breeder.github.io/atlas/ and Fig. 7 showcases the website.

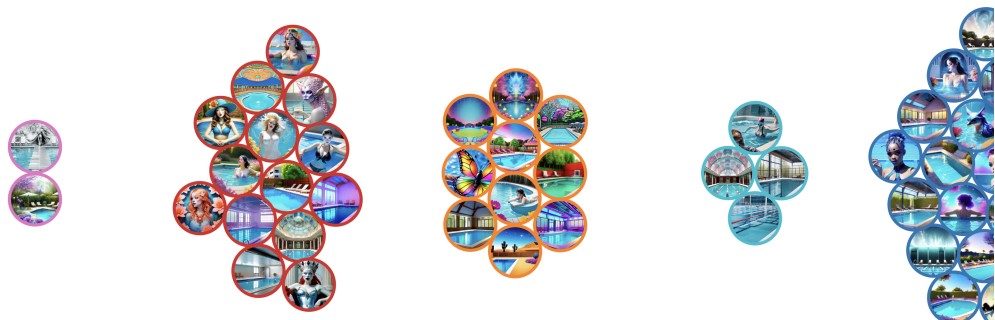

Figure 7: Style Atlas Platform

## E  More Comparisons with Other Datasets

Table 3 features several prominent image synthesis benchmarks. Certain benchmarks focus on single-category image datasets for unconditional image synthesis evaluation, such as face datasets like CelebA [29] and FFHQ [24] (human faces) and MetFaces [23] (face artworks). There are also multi-class datasets like STL-10 [6] and ImageNet [7], but these datasets predominantly consist of photographic images with a limited artwork representation. ArtBench-10 [28] is primarily composed of artwork. However, with only ten classes, it heavily skews toward artwork of North American, European, and East Asian origin and fails to encompass digital and modern art. TWIGMA [5] presents an AI-generated art dataset but suffers from a lack of original generation prompts and a significantly smaller volume of images. The main distinction between our dataset and other AI-generated datasets of similar

scale, such as DiffusionDB [57], lies in the duration over which images were generated. While DiffusionDB covers images generated during a 12-day period in August 2022, STYLEBREEDER extends across a much longer time frame, spanning 18 months from July 2022 to May 2024. This extensive duration provides a significant advantage for in-depth studies into the evolution and dynamics of visual trends, artistic styles, and thematic content. By covering a broader range of temporal variations, our dataset allows for a more detailed analysis of how generative models respond to changing cultural or seasonal influences over time.

Table 3: Summary of Datasets

| Name | Min Resolution | Max Resolution | # Images | Domain |
|---|---|---|---|---|
| MetFaces [23] | (1024, 1024) | (1024, 1024) | 1,336 | Faces (art) |
| STL-10 [6] | (96, 96) | (96, 96) | 13,000 | Objects |
| ArtBench-10 [28] | (32, 32) | (10629, 7437) | 60,000 | Artworks |
| FFHQ [24] | (256, 256) | (1024, 1024) | 70,000 | Faces (Flickr) |
| CelebA [29] | (64, 64) | (1024, 1024) | 202,599 | Faces (celebrities) |
| TWIGMA [5] | $512^2$ | Varied | 800,000 | AI artworks |
| ImageNet [7] | (32, 32) | (256, 256) | 1,431,167 | Objects |
| DiffusionDB [57] | (512, 512) | Varied | 14,000,000 | AI artworks |
| STYLEBREEDER (Ours) | (512, 512) | (1280, 986) | 6,818,217 | AI artworks |

## F  Temporal Trends in Seasonal Content Generation

Our dataset offers potential for deeper exploration of temporal patterns, particularly how seasonal variations influence content generation throughout the year. For instance, a preliminary analysis of the week preceding Halloween (October 24–31) revealed a marked increase in keywords such as *Halloween*, *scary*, *costume*, and *pumpkin* compared to other periods. Further examination of these seasonal trends could yield valuable insights, which we leave as an avenue for future work.

## G  Detailed Analysis on Clustering

We leverage the silhouette score to determine the optimal number of clusters at $k = 10000$ (as can be seen from Tab. 4, we also try 20000 clusters, but it did not offer a significant deviation in silhouette score compared to 10000 clusters). This number of clusters provides the most meaningful and distinct categorization of our data, which is then used for KMeans++ clustering.

Table 4: Number of clusters k and Silhouette Score

| # Clusters | Silhouette Score |
|---|---|
| 50 | 0.032 |
| 100 | 0.043 |
| 500 | 0.054 |
| 1000 | 0.064 |
| 2000 | 0.078 |
| 5000 | 0.087 |
| 10000 | 0.110 |
| 20000 | 0.111 |

We aim to capture artistic styles within each of these clusters using CSD [46] and hypothesize that the majority of the clusters focus on individual artists with unique styles or groups of artists who have similar styles. Further analysis of the number of assigned clusters to unique artist names reveals interesting distribution patterns. We examine the dominance of individual artists within clusters, inspecting the number of clusters where the top contributing artists represented a significant portion of the data points, and observe the following:

- 1551 clusters are dominated by a single artist: This means that in these clusters, over 50% of the data points belong to a single artist, highlighting a strong association between the cluster and that artist's distinct style.
- 2345 clusters are dominated by two artists: This suggests that these clusters capture stylistic similarities between two artists, potentially representing shared influences, overlapping techniques, or a broader stylistic movement encompassing both artists.
- 1467 clusters are dominated by the three artists: This further expands the scope of shared stylistic traits, indicating potential sub-genres or broader artistic trends encompassing a small group of artists.
- 884 clusters are dominated by four artists: This reinforces the trend of clusters capturing shared stylistic qualities among a small group of artists, suggesting the presence of broader artistic movements or schools of thought.

This reveals a fascinating dynamic between individual artistic styles and broader trends. While a significant portion of clusters (1551 or 15.5% of the total 10000 clusters) strongly represent individual artists, a larger portion (almost 52% when considering clusters dominated by the top two, three, or four artists) suggests the presence of shared stylistic traits and broader artistic movements.

## H  Identifying Artistic Influences in Text Prompts

On platforms such as Artbreeder, it is common to use artist names in text prompts to indicate a particular style. We used a popular NER library [54] to extract artist names from the text prompts. The top 18 artist names are provided in Tab. 5. We provided another column in our dataset to indicate potential artist names in the text prompts. Moreover, we provide a Google form on our website if any artist whose name is used would like to opt-out from our dataset.

Table 5: Top 18 artists used in text-prompts.

| Artist Name | Occurrence |
|---|---|
| Tom Bagshaw | 812355 |
| Stanley Artgerm | 547422 |
| Greg Rutkowski | 521464 |
| Daniel F Gerhartz | 430276 |
| WLOP | 389215 |
| Charlie Bowater | 356740 |
| Atey Ghailan | 351338 |
| Andrew Atroshenko | 336390 |
| Rossdraws | 289541 |
| Edouard Bisson | 229375 |
| Alphonse Mucha | 211639 |
| Ilya Kuvshinov | 206632 |
| Mike Mignola | 196128 |
| Pino Daeni | 123757 |
| Krenz Cushart | 120184 |
| Ismail Inceoglu | 107547 |
| Luis Royo | 100998 |
| Guweiz | 99543 |

## I  Copyright Infringement

One of the primary applications of this large-scale dataset of generated content is to understand how much-generated art is a function of an original artist's work. This has a significant impact on the copyright infringement of T2I models. We apply the method from [34], which performs artistic style classification by determining a particular artist's

unique style through a reference dataset of 372 artists' works. This enables us to recognize if identified styles reappear within images in our dataset. Using this successfully identified all 372 artists supported, resulting in 688K images within our dataset. The top artists discovered using this dataset are highlighted in Tab. 6. This demonstrates that our dataset can serve as a valuable resource for scaling up the mentioned method, thereby improving the robustness of artist classification across the 372 artists. Additionally, given the extensive coverage of artists in our dataset, this method could be extended to cover a much broader range of artists, offering valuable insights for addressing copyright infringement issues.

Table 6: Top 10 artist styles detected

| Artist Name | Accuracy | # Samples in STYLEBREEDER |
|---|---|---|
| Alphonse Mucha | 42% | 209357 |
| Ivan Aivazovsky | 39% | 50177 |
| Francis Bacon | 28% | 5565 |
| Claude Monet | 28% | 8210 |
| Vincent Van Gogh | 28% | 14671 |
| Hieronymus Bosch | 27% | 8098 |
| Frank Stella | 23% | 9254 |
| John William Waterhouse | 22% | 60683 |
| Egon Schiele | 21% | 11003 |
| Dan Witz | 21% | 10964 |

## J Datasheet

### MOTIVATION

**For what purpose was the dataset created?** Was there a specific task in mind? Was there a specific gap that needed to be filled? Please provide a description.

As artists worldwide are increasingly leveraging text-to-image diffusion models to create artworks spanning diverse sets of styles, there is a pressing need to uncover and promote unique artistic expressions. This will enable users to discover unique artistic styles, generate personalized content using those styles, and get recommendations about styles they would be interested. STYLEBREEDER project was created to address these questions, and to the best of our knowledge, has the largest userbase (95K) spanning to 6.8M images with 1.8M unique text prompts.

**Who created this dataset (e.g., which team, research group) and on behalf of which entity (e.g., company, institution, organization)?**

This dataset is collected by Matthew Zheng from Virginia Tech and Enis Simsar from ETH Zurich and Joel Simon from Artbreeder.

**What support was needed to make this dataset?** (e.g.who funded the creation of the dataset? If there is an associated grant, provide the name of the grantor and the grant name and number, or if it was supported by a company or government agency, give those details.)

No specific funding was used.

**Any other comments?**

No.

### COMPOSITION

**What do the instances that comprise the dataset represent (e.g., documents, photos, people, countries)?** Are there multiple types of instances (e.g., movies, users, and ratings; people and interactions between them; nodes and edges)? Please provide a description.

Each instance consists of an image generated by a text-to-image diffusion model (such as SD-XL or ControlNet) and the original text prompt used for generating the image, as well as original metadata and expanded features created by us. The metadata originally collected from the Artbreeder website are Positive Prompt, Negative Prompt, anonymized ImageID, anonymized UserID, Timestamp, Image Size (height, width), and model-related hyperparameters, including Model Type, Seed, Step, and CFG Scale. We also offer further metadata like Cluster ID, along with scores for Prompt NSFW, Image NSFW, Toxicity, Insult, Threat, and Identity Attack computed with [15, 16]. We also provide potential artist names that are used in the text prompt, such as "Ilya Kuvshinov".

**How many instances are there in total (of each type, if appropriate)?**

6,818,217 images were created by 95,479 unique users, with 1.8M unique text prompts.

**Does the dataset contain all possible instances or is it a sample (not necessarily random) of instances from a larger set?** If the dataset is a sample, then what is the larger set? Is the sample representative of the larger set (e.g., geographic coverage)? If so, please describe how this representativeness was validated/verified. If it is not representative of the larger set, please describe why not (e.g., to cover a more diverse range of instances, because instances were withheld or unavailable).

No tests were run to determine the representativeness of the dataset. However, our dataset consists of all publicly generated images on the Artbreeder Platform's Collage tool from July 2022 to May 2024. We provide a 2M version which includes all data and the original images, hosted on a publicly available platform: https://huggingface.co/datasets/stylebreeder/stylebreeder. Due to resource constraints, we provide a script to download the full data on our website: https://huggingface.co/datasets/stylebreeder/stylebreeder. The full data is divided into a 5M version and a 6.8M version. The 5M version includes images filtered by LLavaGuard [17] to remove potentially unsafe content. The 6.8M version includes

both safe and unsafe text prompts and will be shared with researchers to promote research in safety and ethics, and will be shared upon receiving information regarding intended usage via this Google Form: https://forms.gle/FtQ44igxpbgz6LHe8.

**What data does each instance consist of?** "Raw" data (e.g., unprocessed text or images) or features? In either case, please provide a description.
Each instance contains a unique image ID, corresponding meta, and expanded features, including Positive Prompt, Negative Prompt, anonymized UserID, Timestamp, Image Size (height, width), and model-related hyperparameters, including Model Type, Seed, Step, and CFG Scale, Cluster ID, Prompt NSFW, Image NSFW, Toxicity, Insult, Threat, Identity Attack scores, Artist names. We also provide the original image files along with 2M subset https://huggingface.co/datasets/stylebreeder/stylebreeder, and a script to download the 5M version.

**Is there a label or target associated with each instance?** If so, please provide a description.
For each image, we provide an extended set of labels, including text prompt, model parameters, and other features as discussed above.

**Is any information missing from individual instances?** If so, please provide a description, explaining why this information is missing (e.g., because it was unavailable). This does not include intentionally removed information, but might include, e.g., redacted text.
The public version of the dataset is filtered by LLavaGuard [17] to remove potentially unsafe content.

**Are relationships between individual instances made explicit (e.g., users' movie ratings, social network links)?** If so, please describe how these relationships are made explicit.
Each image is connected to an anonymized user ID. By filtering the dataset by user ID, one can infer the relationship between different samples where the same user creates those samples.

**Are there recommended data splits (e.g., training, development/validation, testing)?** If so, please provide a description of these splits, explaining the the rationale behind them.
We do not provide any recommended splits.

**Are there any errors, sources of noise, or redundancies in the dataset?** If so, please provide a description.
No, all images and related metadata are collected directly from the Artbreeder platform.

**Is the dataset self-contained, or does it link to or otherwise rely on external resources (e.g., websites, tweets, other datasets)?** If it links to or relies on external resources, a) are there guarantees that they will exist, and remain constant, over time; b) are there official archival versions of the complete dataset (i.e., including the external resources as they existed at the time the dataset was created); c) are there any restrictions (e.g., licenses, fees) associated with any of the external resources that might apply to a future user? Please provide descriptions of all external resources and any restrictions associated with them, as well as links or other access points, as appropriate.
We provide a 2M version which includes all data and the original images, hosted on a publicly available platform: https://huggingface.co/datasets/stylebreeder/stylebreeder.

**Does the dataset contain data that might be considered confidential (e.g., data that is protected by legal privilege or by doctor-patient confidentiality, data that includes the content of individuals' non-public communications)?** If so, please provide a description.
The dataset is collected from users' publicly available content; however, since there is no control over what type of text prompts users can input, it is possible that some confidential information might be included in the text prompts; however, we expect such cases to be rare.

**Does the dataset contain data that, if viewed directly, might be offensive, insulting, threatening, or might otherwise cause anxiety?** If so, please describe

why.
Similar to other datasets with AI-generated images like DiffusionDB and TWIGMA, ours contains a substantial amount of potentially NSFW content regarding images and text prompts. This observation correlates with recent studies highlighting a significant increase in NSFW content generation by online communities [38, 43]. We provide an analysis on NSFW and other features such as Toxicity, Severe Toxicity, Identity Attack, Obscene, Insult, and Threat in Fig. 3 and Sec. 3.4 in the main paper. Moreover, to help researchers to filter out potentially unsafe content we provide these scores in our dataset.

**Does the dataset relate to people?** If not, you may skip the remaining questions in this section.
We share anonymized user IDs, and text prompts may have celebrity or artist names in the text prompts.

**Does the dataset identify any subpopulations (e.g., by age, gender)?** If so, please describe how these subpopulations are identified and provide a description of their respective distributions within the dataset.
No.

**Is it possible to identify individuals (i.e., one or more natural persons), either directly or indirectly (i.e., in combination with other data) from the dataset?** If so, please describe how.
Text prompts may have celebrity or artist names in them. We provide a Google form for reporting harmful or inappropriate images and prompts, as well as allowing artists to opt-out in case their names are used in the text prompts.

**Does the dataset contain data that might be considered sensitive in any way (e.g., data that reveals racial or ethnic origins, sexual orientations, religious beliefs, political opinions or union memberships, or locations; financial or health data; biometric or genetic data; forms of government identification, such as social security numbers; criminal history)?** If so, please provide a description.
We do not collect any user-related data except anonymized user IDs.

**Any other comments?**
No.

---

## COLLECTION

**How was the data associated with each instance acquired?** Was the data directly observable (e.g., raw text, movie ratings), reported by subjects (e.g., survey responses), or indirectly inferred/derived from other data (e.g., part-of-speech tags, model-based guesses for age or language)? If data was reported by subjects or indirectly inferred/derived from other data, was the data validated/verified? If so, please describe how.
The images, prompt, and other metadata, including Positive Prompt, Negative Prompt, anonymized ImageID, anonymized UserID, Timestamp, and Image Size (height, width), and model-related hyperparameters, including Model Type, Seed, Step, and CFG Scale, are collected from Artbreeder website. Moreover, we derived additional features such as Cluster ID, Prompt NSFW, Image NSFW, Toxicity, Insult, Threat, Identity Attack scores, Artist names [15, 16, 55].

**Over what timeframe was the data collected?** Does this timeframe match the creation timeframe of the data associated with the instances (e.g., recent crawl of old news articles)? If not, please describe the timeframe in which the data associated with the instances was created. Finally, list when the dataset was first published.
All data was collected from April-May 2024, The images generated between July 2022-May 2024. The dataset was first published with this NeurIPS submission on June 10, 2024.

**What mechanisms or procedures were used to collect the data (e.g., hardware apparatus or sensor, manual human curation, software program, software API)?** How were these mechanisms or procedures validated?
The image IDs and original metadata are directly pulled from the database by one of the

authors, Joel Simon, owner of the Artbreeder platform. A Python script was used to scrape the images and infer additional features such as Cluster ID or NSFW scores.

**What was the resource cost of collecting the data?** (e.g. what were the required computational resources, and the associated financial costs, and energy consumption - estimate the carbon footprint. See Strubell *et al.* for approaches in this area.)

Artbreeder provided the metadata and image IDs directly via their database. Collecting the corresponding images took a total of 78 hours, estimated bandwidth Costs: \$315, storage Costs: \$80.5, and Energy Costs: \$3.9 (assuming an electricity cost of \$0.10 per kWh to run a server for 78 hours).

**If the dataset is a sample from a larger set, what was the sampling strategy (e.g., deterministic, probabilistic with specific sampling probabilities)?**

We collected all data from Artbreeder's Collage tool from July 2022 to May 2024, so this is not a subset.

**Who was involved in the data collection process (e.g., students, crowdworkers, contractors) and how were they compensated (e.g., how much were crowdworkers paid)?**

Students collected the data as a part of their graduate research assistantship at Virginia Tech or ETH Zurich.

**Were any ethical review processes conducted (e.g., by an institutional review board)?** If so, please provide a description of these review processes, including the outcomes, as well as a link or other access point to any supporting documentation.

No ethical review process was conducted.

**Does the dataset relate to people?** If not, you may skip the remainder of the questions in this section.

Our dataset relates to people in 2 ways: 1) We share anonymized user IDs for users who generate the images. 2) Some text prompts may have celebrity or artist names in the prompt.

**Did you collect the data from the individuals in question directly, or obtain it via third parties or other sources (e.g., websites)?**

We collected the data from the Artbreeder website.

**Were the individuals in question notified about the data collection?** If so, please describe (or show with screenshots or other information) how notice was provided, and provide a link or other access point to, or otherwise reproduce, the exact language of the notification itself.

No, according to Artbreeder's contract, every user accepts the agreement that their data is licensed as CC0 (Public Domain). The exact language can be seen from: `https://www.artbreeder.com/terms.pdf`.

**Did the individuals in question consent to the collection and use of their data?** If so, please describe (or show with screenshots or other information) how consent was requested and provided, and provide a link or other access point to, or otherwise reproduce, the exact language to which the individuals consented.

In Artbreeder's contract, every user accepts the agreement that their data is licensed as CC0 (Public Domain). Hence, we did not ask for additional consent.

**If consent was obtained, were the consenting individuals provided with a mechanism to revoke their consent in the future or for certain uses?** If so, please provide a description, as well as a link or other access point to the mechanism (if appropriate)

Users will have the option to report any harmful, unsafe and sensitive content via our Google form on our website: `http://stylebreeder.github.io`. Also, artists whose names are used in the text prompts can opt-out via the same form.

**Has an analysis of the potential impact of the dataset and its use on data subjects (e.g., a data protection impact analysis) been conducted?** If so, please provide a description of this analysis, including the outcomes, as well as a link or other access point to any supporting documentation.

No data protection impact analysis has been conducted.

**Any other comments?**
No.

---

## PREPROCESSING / CLEANING / LABELING

---

**Was any preprocessing/cleaning/labeling of the data done(e.g.,discretization or bucketing, tokenization, part-of-speech tagging, SIFT feature extraction, removal of instances, processing of missing values)?** If so, please provide a description. If not, you may skip the remainder of the questions in this section.
We used a library to infer various scores such as NSFW, Toxicity, Insult, Threat, and Identity Attack scores using Detoxify library [15]. Image NSFW scores are computed using NSFW-Detector library [16]. To infer artist names, we used Named Entity Recognition via [55].

**Was the "raw" data saved in addition to the preprocessed/cleaned/labeled data (e.g., to support unanticipated future uses)?** If so, please provide a link or other access point to the "raw" data.
Yes, raw data is saved.

**Is the software used to preprocess/clean/label the instances available?** If so, please provide a link or other access point.
All of our data collection and preprocessing code is available on our website: `http://stylebreeder.github.io`

**Any other comments?**
No.

---

## USES

---

**Has the dataset been used for any tasks already?** If so, please provide a description.

In our paper, we showcased a variety of tasks; 1) discovering diverse styles using clustering based on style representations, 2) personalized content generation on these styles with LoRA and similar methods 3) style recommendation.

**Is there a repository that links to any or all papers or systems that use the dataset?** If so, please provide a link or other access point.
This is the first paper using this dataset, however if there are other papers or systems use this dataset in the future, we will include a link in our website.

**What (other) tasks could the dataset be used for?**
As discussed in our paper, our dataset offers numerous avenues for further exploration, such as refining the effectiveness of text prompts through iterative adjustments, studying trends in art over time, recommending styles based on both image and textual content, developing a search system for the generated images, and exploring explainable creativity.

**Is there anything about the composition of the dataset or the way it was collected and preprocessed/cleaned/labeled that might impact future uses?** For example, is there anything that a future user might need to know to avoid uses that could result in unfair treatment of individuals or groups (e.g., stereotyping, quality of service issues) or other undesirable harms (e.g., financial harms, legal risks) If so, please provide a description. Is there anything a future user could do to mitigate these undesirable harms?
Since we collected full data instead of a subset, we do not expect any unfair treatment of individuals or groups. Moreover, since the dataset is already in the public domain, we do not expect any legal or financial risks. However, as discussed in our paper, some of the images and text prompts reflect NSFW content, and researchers should use the associated scores we provided to filter out unsafe data based on their tasks.

**Are there tasks for which the dataset should not be used?** If so, please provide a description.
All tasks utilizing this dataset should follow CC0 Public domain licensing rules.

**Any other comments?**
No.

---

## DISTRIBUTION

**Will the dataset be distributed to third parties outside of the entity (e.g., company, institution, organization) on behalf of which the dataset was created?** If so, please provide a description.
Yes, the dataset is available on the internet.

**How will the dataset will be distributed (e.g., tarball on website, API, GitHub)?** Does the dataset have a digital object identifier (DOI)?
We share the dataset, code, and other information on our website: `http://stylebreeder.github.io`.

**When will the dataset be distributed?**
The dataset is released on June 12, 2024.

**Will the dataset be distributed under a copyright or other intellectual property (IP) license, and/or under applicable terms of use (ToU)?** If so, please describe this license and/or ToU, and provide a link or other access point to, or otherwise reproduce, any relevant licensing terms or ToU, as well as any fees associated with these restrictions.
All images generated on Artbreeder platform are under the CC0 Public Domain License, therefore our images, text prompts and other metadata follows the same rules.

**Have any third parties imposed IP-based or other restrictions on the data associated with the instances?** If so, please describe these restrictions, and provide a link or other access point to, or otherwise reproduce, any relevant licensing terms, as well as any fees associated with these restrictions.
All images generated on the Artbreeder platform are under the CC0 Public Domain License.

**Do any export controls or other regulatory restrictions apply to the dataset or to individual instances?** If so, please describe these restrictions, and provide a link or other access point to, or otherwise reproduce, any supporting documentation.
No.

**Any other comments?**
No.

---

## MAINTENANCE

**Who is supporting/hosting/maintaining the dataset?**
The authors of this paper are responsible for supporting, hosting, and maintaining the dataset. We also provide a Google form for users to report any potential issue about the dataset.

**How can the owner/curator/manager of the dataset be contacted (e.g., email address)?**
The contact information is listed on our website: `http://stylebreeder.github.io`. We also provide a Google form for users to report any potential issue about the dataset.

**Is there an erratum?** If so, please provide a link or other access point.
There is no erratum with our initial release, but we will update our website if we have any errors in the future.

**Will the dataset be updated (e.g., to correct labeling errors, add new instances, delete instances)?** If so, please describe how often, by whom, and how updates will be communicated to users (e.g., mailing list, GitHub)?
Authors of this paper will monitor the Google form where users can report any potential issues. We will announce the updates on our website: `http://stylebreeder.github.io`.

**If the dataset relates to people, are there applicable limits on the retention of the data associated with the instances (e.g., were individuals in question told that their data would be retained for a fixed period of time and then deleted)?**

Users can contact us from the Google form and report any potentially harmful or sensitive information for removal.

Older versions of the dataset will be hosted.

Other researchers are welcome to extend or contribute to our dataset. They can either send a pull request to our GitHub repository (`http://github.com/stylebreeder`) or contact authors on Google form.

No.

