# OpenReview forum: "Stylebreeder: Exploring and Democratizing Artistic Styles through Text-to-Image Models"
_NeurIPS.cc/2024/Datasets_and_Benchmarks_Track — NeurIPS 2024 Track Datasets and Benchmarks Poster_

### Official Review · Reviewer_yd7g · 2024-07-02
**Review for Stylebreeder**

**Rating:** 6
**Confidence:** 4
**Correctness:** Yes, the dataset is constructed in a …
**Clarity:** The paper is well-written!

**Review:**

Strengths:

 - The dataset is large-scale and contains rich annotations (e.g., better metadata compared to other datasets / databases such as DiffusionDB) - this makes the dataset useful for downstream analysis.

- The paper is well-written and easy to follow.

- Well-executed applications for personalized image generation and recommendations. I believe these are some practical creative applications for leveraging the dataset for enabling users to generate more personalized artworks.



Weaknesses:

- While the dataset is useful to understand the landscape of user-generated art, the paper lacks in useful analysis. While personalized image generation is a practical application and the authors execute that part well, they do not have relevant analysis on what the clusters of image styles represent. The authors identify 1000 clusters - but do not provide significant insights on : (i) how these 1000 clusters are selected using the CSD model; (ii) what do those clusters represent? Are they heavily biased towards a particular artist name in each cluster.  Also are there clusters where the artist's name is not represented well, but it contains qualitative aspects of an artist due to the other keywords?

- The paper should contain more discussion about the comparison to DiffusionDB considering it is larger and also contain prompt. I think from a practical perspective one can also use CSD with DiffusionDB to obtain clusters of different styles, which can be used towards personal image generation. While this paper contains images from more models, does it really contribute towards more diverse personalized image generation?

- The authors are suggested to add some more discussion on the impact of this dataset in the realm of copyright infringements. I think this is one of the primary applications of this dataset - to understand how much generated art is a function of the original artist's work in practice. Maybe one can apply the method in https://arxiv.org/abs/2404.08030 on their dataset to understand copyright infringements better.

Overall, the dataset is useful for understanding the landscape of user-generated art, but the paper lacks in strong analysis on top of the dataset. I am happy to revisit my scores, if the authors can provide clarifications on the Weaknesses.

**Strengths:**

Check strengths above.

**Additional Feedback:**

I would suggest the authors to iterate on the Weaknesses to make the paper better. As mentioned I am happy to revisit my scores if the concerns are addressed.

**Documentation:**

Yes!

**Limitations:**

Yes, the authors have addressed potential negative societal impact of their work.

**Opportunities For Improvement:**

Check the Weaknesses section.

**Relation To Prior Work:**

Yes, but I have provided a suggestion to present the work better in the context of DiffusionDB.

**Summary And Contributions:**

This paper introduces the Stylebreeder dataset, collected from the Artbreeder website, which includes large-scale stylistic images along with their associated metadata, and NSFW scores. This richly annotated dataset is used for experiments in diverse artistic styles, personalized image generation, and style-based recommendation. The authors analyze the dataset, conduct experiments on style identification, clustering styles, and discuss the limitations and societal impact of AI in art generation.

---

> ### Author Rebuttal · Authors · 2024-08-16
>
> # Rebuttal (1/2)
>
> > R3-Q1: While the dataset is useful to understand the landscape of user-generated art, the paper lacks in useful analysis. While personalized image generation is a practical application and the authors execute that part well, they do not have relevant analysis on what the clusters of image styles represent. The authors identify 1000 clusters - but do not provide significant insights on : (i) how these 1000 clusters are selected using the CSD model
>
> We apologize for the typo; our dataset actually consists of 10000 clusters, as verified by the 'cluster_id' column. To determine the optimal number of clusters, we applied the silhouette score method across sizes ranging from 50 to 20000. This method assesses cohesion within a cluster versus separation from other clusters. Our findings revealed that 10000 clusters achieve the highest silhouette score, suggesting optimal internal similarity and external separation. This optimal clustering configuration, used in our KMeans++ clustering, ensures the most meaningful and distinct data categorization. For detailed scores, please refer to the table shared in R1-Q2.
>
> > R3-Q2: (ii) what do those clusters represent? Are they heavily biased towards a particular artist name in each cluster. Also are there clusters where the artist's name is not represented well, but it contains qualitative aspects of an artist due to the other keywords?
>
> Since we are aiming to capture artistic styles within these clusters using the CSD metric, we expect the majority of the clusters to focus on individual artists with unique styles, or groups of artists who have similar styles. Indeed, further analysis on the number of assigned clusters to unique artist names reveal interesting distribution patterns. We examined the dominance of individual artists within clusters, looking at the number of clusters where the top contributing artists represented a significant portion of the data points. Here's what we found:
>
> - 1551 clusters are dominated by a single artist: This means that in these clusters, over 50% of the data points belong to a single artist, highlighting a strong association between the cluster and that artist's distinct style.
> - 2345 clusters are dominated by two artists: This suggests that these clusters capture stylistic similarities between two artists, potentially representing shared influences, overlapping techniques, or a broader stylistic movement encompassing both artists.
> - 1467 clusters are dominated by the three artists: This further expands the scope of shared stylistic traits, indicating potential sub-genres or broader artistic trends encompassing a small group of artists.
> - 884 clusters are dominated by four artists: This reinforces the trend of clusters capturing shared stylistic qualities among a small group of artists, suggesting the presence of broader artistic movements or schools of thought.
>
> These numbers reveal a fascinating dynamic between individual artistic styles and broader trends. While a significant portion of clusters (1551 or 15.5% of the total 10000 clusters) strongly represent individual artists, a larger portion (almost 52% when considering clusters dominated by the top two, three, or four artists) suggests the presence of shared stylistic traits and broader artistic movements.
>
> > R3-Q3: The paper should contain more discussion about the comparison to DiffusionDB considering it is larger and also contain prompt. I think from a practical perspective one can also use CSD with DiffusionDB to obtain clusters of different styles, which can be used towards personal image generation.
>
> The main distinction between our dataset and DiffusionDB lies in the duration over which the images were generated. While DiffusionDB's dataset covers a brief period of just 12 days in August 2022, our dataset extends across a much longer timeframe, spanning 18 months from July 2022 to May 2024. This extensive duration provides a significant advantage for in-depth studies into the evolution and dynamics of visual trends, artistic styles, and thematic content. By covering a broader range of temporal variations, our dataset allows for a more detailed analysis of how generative models respond to changing cultural, or seasonal influences over time. For instance, our preliminary investigation into the period around Halloween, specifically the seven days leading up to October 31, identified keywords like `Halloween`, `scary`, `costume`, and `pumpkin` much more frequently compared to other weeks. We plan to enrich the camera-ready version of our discussion by exploring deeper into the temporal aspects of our dataset, particularly how seasonal changes influence the content generated during different times of the year.
>
> > R3-Q4: While this paper contains images from more models, does it really contribute towards more diverse personalized image generation?
>
> Our dataset offers a variety of T2I models such as SD, SD-XL or ControlNet, supporting resolutions ranging from 512 × 512 to 1280 × 896. These models differ in their capabilities and the quality of their generated outputs, with recent models often supporting higher resolutions that deliver finer details and more complex visuals. This variety provides a valuable opportunity to explore differences between models, such as variations in artistic expression, the nuances in image quality, and potential biases inherent in each model. Analyzing these distinctions can lead to deeper insights into how different generative models perform and interact with user inputs. We will include a discussion about this to the camera ready version.

---

> > ### Author Rebuttal · Authors · 2024-08-16
> >
> > # Rebuttal (2/2)
> >
> > > R3-Q5: The authors are suggested to add some more discussion on the impact of this dataset in the realm of copyright infringements. I think this is one of the primary applications of this dataset - to understand how much generated art is a function of the original artist's work in practice. Maybe one can apply the method in https://arxiv.org/abs/2404.08030 on their dataset to understand copyright infringements better.
> >
> > We applied the method from the suggested paper to our dataset and successfully identified all 372 artists supported by that method, resulting in 688K images within our dataset. The top artists discovered using this dataset are highlighted below. This demonstrates that our dataset can serve as a valuable resource for scaling up the mentioned method, thereby improving the robustness of artist classification across the 372 artists. Additionally, given the extensive coverage of artists in our dataset, this method could be extended to cover a much broader range of artists, offering valuable insights for addressing copyright infringement issues. We will expand on this discussion in the camera-ready version.
> >
> > **Their Model on Our Dataset (Only Inference) Over 5K Samples Each**
> >
> > | Artist                  | Accuracy | Number of Samples in our dataset |
> > |-------------------------|----------|----------------------------------|
> > | Alphonse Mucha          | 42%      | 209357                          |
> > | Ivan Aivazovsky          | 39%      | 50177                           |
> > | Francis Bacon           | 28%      | 5565                            |
> > | Claude Monet            | 28%      | 8210                            |
> > | Vincent Van Gogh        | 28%      | 14671                           |
> > | Hieronymus Bosch        | 27%      | 8098                            |
> > | Frank Stella            | 23%      | 9254                            |
> > | John William Waterhouse | 22%      | 60683                           |
> > | Egon Schiele            | 21%      | 11003                           |
> > | Dan Witz                | 21%      | 10964                           |

---

> > > ### Comment · Reviewer_yd7g · 2024-08-28
> > > **Response**
> > >
> > > Thanks for the detailed rebuttal. I have increased the score.

---

> > > > ### Author Rebuttal · Authors · 2024-08-28
> > > >
> > > > Thank you for your comment. We will update our camera-ready version to incorporate your feedback.

---

> ### Author Response · Authors · 2024-08-26
>
> We appreciate your time and effort in reviewing our submission. If you have any feedback on our rebuttal or additional comments that need addressing, we would be happy to provide more details.

---

### Official Review · Reviewer_o4hr · 2024-07-24
**A large-scale art-style dataset of images generated by text-to-image models**

**Rating:** 7
**Confidence:** 4
**Correctness:** The dataset is constructed in a sound…
**Clarity:** This paper is well-written and easy t…

**Review:**

Quality: This paper proposes a dataset consisting of 6.8M images generated by text-to-image models, mainly Stable Diffusion models. The dataset can be used to analyze the textual expression of art styles with the user prompts. With the dataset, the authors also provide LoRA modules for identified styles so that users can customize the art style easily.

Clarity: This paper is well-written and easy to follow. The related work in this field is well discussed.

Originality: The originality of the newly proposed dataset is good.

Significance: This work contributes a large-scale dataset of images and prompts of text-to-image models for art style analysis and customization, which can benefit both the academic and creative usage community.

**Strengths:**

1. The curated dataset contains images and prompts of multiple text-to-image models in a time period of 1.5 years, which is collected from a platform with a large number of users and makes this dataset valuable for followup research.

2. The authors analyzed the art styles based on the collected dataset and identified clear clusterings of styles, showing the value of the dataset.

3. The authors provided LoRAs for multiple identified styles from the dataset and released them at a public platform for easier access of customization with the styles.

**Additional Feedback:**

No additional feedback.

**Documentation:**

The paper provides enough details for the proposed dataset. The contributed dataset and platform are released and available as well.

**Ethics:**

There are no or minor ethical concerns with the submission.

**Limitations:**

The authors adequately discussed the limitations of their work.

**Opportunities For Improvement:**

1. Is there any bias from the Artbreeder platform? What's the motivation of users for uploading images and prompts to this platform?

2. How to guarantee the quality of the collected images and prompts? Is there any data cleaning or filtering during the curation of the dataset?

3. There is a disparity between the number of images and prompts. Does that indicate that multiple prompts correspond to the same image and the same user uploads the images to the website?

4. It looks like the images of the Stable Diffusion variants take up the majority of the dataset, could the authors briefly discuss the possible reasons why users prefer uploading images from Stable Diffusion, instead of other models such as Midjourney DALL-E or others?

**Relation To Prior Work:**

This paper discussed prior work in detail.

**Summary And Contributions:**

This paper contributes a dataset named Stylebreeder for the creative usage of text-to-image models. The dataset consists of 6.8M images and 1.8M prompts generated by 95K users. By analyzing the images, the authors identified and analyzed novel, user-generated artistic styles, which can be undocumented new expressions to describe artistic styles in the era of text-to-image models. The dataset can provide support for multiple tasks including identifying various artistic styles, creating customized content, and recommending styles tailored to user preferences. This paper also provides a web-based platform named Style Atlas for users to download pre-trained style LoRAs for personalized content generation.

---

> ### Author Rebuttal · Authors · 2024-08-16
>
> > R2-Q1: Is there any bias from the Artbreeder platform? What's the motivation of users for uploading images and prompts to this platform?
>
> Artbreeder provides users with a platform to create images using text prompts, offering controls over various settings such as the strength (guidance scale) of the text's influence on the generated image, seed values, model type, and other hyperparameters. Since its rise in popularity within the artistic community in 2018, Artbreeder has become known for its bias towards generating artistic images. This predisposition towards artistic styles is a primary reason we concentrated our focus in this area. We will provide a more detailed discussion on this topic in the camera-ready version of our paper.
>
> > R2-Q2: How to guarantee the quality of the collected images and prompts? Is there any data cleaning or filtering during the curation of the dataset?
>
> Our dataset includes all images generated on the Artbreeder platform from July 2022 to May 2024. We excluded any corrupted images and those generated from empty text prompts. To enhance the dataset, we appended additional properties such as Prompt NSFW, Image NSFW, and scores for Toxicity, Severe Toxicity, Identity Attack, Obscenity, Insult, and Threat. These properties are included within our dataset to provide comprehensive insights into the content generated. In addition to the properties above, we included artist names, and cluster assignments for future research.
>
> > R2-Q3: There is a disparity between the number of images and prompts. Does that indicate that multiple prompts correspond to the same image and the same user uploads the images to the website?
>
> Our dataset comprises 1.8 million unique text prompts that correspond to 6.8 million images. The disparity arises because different images can be generated from the same text prompt when varying parameters such as the guidance scale, seed values, and generation steps. This variability demonstrates another valuable aspect of our dataset. By examining how these different configurations affect the resulting images, researchers can gain deeper insights into the model's behavior and its sensitivity to these parameters. This analysis enables a deeper understanding of how subtle changes in input or settings can significantly alter the characteristics of generated images, providing valuable perspectives on the underlying generative processes. We will elaborate this further in the camera ready version.
>
> > R2-Q4: It looks like the images of the Stable Diffusion variants take up the majority of the dataset, could the authors briefly discuss the possible reasons why users prefer uploading images from Stable Diffusion, instead of other models such as Midjourney DALL-E or others?
>
> Stable Diffusion and its variants are predominantly used on the Artbreeder platform primarily due to their open-source nature, in contrast to platforms like DALL-E and Midjourney, which are not open-source. This accessibility is a key reason why images from Stable Diffusion variants feature prominently in our dataset. It’s important to clarify that users do not upload images to the Artbreeder platform; instead, they generate images directly on the platform using various controls such as text prompts, hyperparameters, and other settings. This method of image creation allows users to actively engage with and manipulate the generative process.

---

> > ### Comment · Reviewer_o4hr · 2024-08-26
> >
> > Thanks for the detailed rebuttal from the authors. I have several follow-up questions regarding this paper.
> > 1. Is there any recognizable bias that authors notice during the image collection? It will be helpful to note or discuss that in the paper to benefit the further usage of this dataset.
> > 2. Could the authors further elaborate how the platform works regarding image collection? Does the platform integrate the models so that users use Artbreeder platform to generate images directly?
> > 3. Is there any platform that allows users to upload their images generated by text-to-image models? Does it hurt the results if the majority of images come from the stable diffusion?

---

> > > ### Author Rebuttal · Authors · 2024-08-26
> > >
> > > We would like to thank to R2 for valuable feedback. Our responses are below:
> > >
> > > > R2-Q5: 1. Is there any recognizable bias that authors notice during the image collection? It will be helpful to note or discuss that in the paper to benefit the further usage of this dataset.
> > >
> > > Yes, we observed that certain artist names appear more frequently in text prompts than others (in other words, users are more biased to use certain artist names). The table below highlights the distribution of these artists based on the number of prompts they are mentioned in:
> > >
> > > | Artist Name          | Occurrence |
> > > |----------------------|------------|
> > > | Tom Bagshaw          | 812355     |
> > > | Stanley Artgerm      | 547422     |
> > > | Greg Rutkowski       | 521464     |
> > > | Daniel F Gerhartz    | 430276     |
> > > | WLOP                 | 389215     |
> > > | Charlie Bowater      | 356740     |
> > > | Atey Ghailan         | 351338     |
> > > | Andrew Atroshenko    | 336390     |
> > > | Rossdraws            | 289541     |
> > > | Edouard Bisson       | 229375     |
> > > | Alphonse Mucha       | 211639     |
> > > | Ilya Kuvshinov       | 206632     |
> > > | Mike Mignola         | 196128     |
> > > | Pino Daeni           | 123757     |
> > > | Krenz Cushart        | 120184     |
> > > | Ismail Inceoglu      | 107547     |
> > > | Luis Royo            | 100998     |
> > > | Guweiz               | 99543      |
> > >
> > >
> > > For future research, it would be valuable to investigate the factors contributing to the popularity of certain artists over others and to examine if this popularity influences the resulting image output.
> > >
> > >
> > > > R2-Q6: 2. Could the authors further elaborate how the platform works regarding image collection? Does the platform integrate the models so that users use Artbreeder platform to generate images directly?
> > >
> > > Yes, Artbreeder integrates Stable Diffusion and its variants (e.g., SDXL, ControlNet) into its backend, enabling users to interact with these models directly through the user interface. We have collected our dataset from the images generated by users via this interface.
> > >
> > > > R2-Q7: 3. Is there any platform that allows users to upload their images generated by text-to-image models? Does it hurt the results if the majority of images come from the stable diffusion?
> > >
> > > Artbreeder also provides functionality for users to upload and alter images using inpainting techniques. However, our dataset exclusively contains examples where users generate images from scratch, as our research concentrates on text prompts and the images they generate.

---

> > > > ### Comment · Reviewer_o4hr · 2024-08-29
> > > >
> > > > Thanks for the detailed discussion with the authors. My concerns have been addressed and I would like to keep my positive rating. Please make sure to include the discussion during the rebuttal in the paper.

---

> > > > > ### Author Rebuttal · Authors · 2024-08-29
> > > > >
> > > > > Thank you for your comment. We will update our camera-ready version to incorporate the discussions during the rebuttal.

---

### Official Review · Reviewer_WApe · 2024-07-24
**Stylebreeder: Exploring and Democratizing Artistic Styles through Text-to-Image Models**

**Rating:** 8
**Confidence:** 5
**Correctness:** yes
**Clarity:** yes

**Review:**

Strength:
1. Extensive Dataset: It provides a comprehensive dataset from Artbreeder with millions of user-generated images and styles under a CC0 license, fostering further research.
2. Web-Based Platform: The release of the Style Atlas platform enables public access.
3. It introduces a recommendation system that aligns style suggestions with individual preferences, enhancing the relevance of artistic exploration.

Weakness:
1. The dataset assembled in this paper, named STYLEBREEDER, demonstrates only a slight advancement when juxtaposed with DiffusionDB, and it notably falls short in terms of sheer quantity compared to DiffusionDB. Although the author highlights that STYLEBREEDER has assimilated images derived from a variety of models, it fails to provide a comprehensive analysis of how images produced by these differing models influence a spectrum of tasks. Consequently, the innovative edge of this dataset remains uncertain.

2. The K-Means++ algorithm is employed to cluster the data into 1,000 groups, but it is not clear how the number 1,000 was determined. Furthermore, the impact of different K settings on the characteristics of the dataset should be subject to further analysis.

**Strengths:**

1. Extensive Dataset: It provides a comprehensive dataset from Artbreeder with millions of user-generated images and styles under a CC0 license, fostering further research.
2. Web-Based Platform: The release of the Style Atlas platform enables public access.
3. It introduces a recommendation system that aligns style suggestions with individual preferences, enhancing the relevance of artistic exploration.

**Additional Feedback:**

The authors should provide further elaboration on the aforementioned question.

**Documentation:**

yes

**Limitations:**

yes

**Opportunities For Improvement:**

1. The dataset assembled in this paper, named STYLEBREEDER, demonstrates only a slight advancement when juxtaposed with DiffusionDB, and it notably falls short in terms of sheer quantity compared to DiffusionDB. Although the author highlights that STYLEBREEDER has assimilated images derived from a variety of models, it fails to provide a comprehensive analysis of how images produced by these differing models influence a spectrum of tasks. Consequently, the innovative edge of this dataset remains uncertain.

2. The K-Means++ algorithm is employed to cluster the data into 1,000 groups, but it is not clear how the number 1,000 was determined. Furthermore, the impact of different K settings on the characteristics of the dataset should be subject to further analysis.

**Relation To Prior Work:**

yes

**Summary And Contributions:**

This paper presents a extensive dataset capturing millions of user-generated images and styles, and introduces a novel platform called Style Atlas that offers personalized content generation and recommendations based on LoRA.

---

> ### Author Rebuttal · Authors · 2024-08-16
>
> > R1-Q1: The dataset assembled in this paper, named STYLEBREEDER, demonstrates only a slight advancement when juxtaposed with DiffusionDB, and it notably falls short in terms of sheer quantity compared to DiffusionDB. Although the author highlights that STYLEBREEDER has assimilated images derived from a variety of models, it fails to provide a comprehensive analysis of how images produced by these differing models influence a spectrum of tasks. Consequently, the innovative edge of this dataset remains uncertain.
>
> The main distinction between our dataset and DiffusionDB lies in the duration over which the images were generated. While DiffusionDB dataset covers a brief period of just 12 days in August 2022, our dataset extends across a much longer time frame, spanning 18 months from July 2022 to May 2024. This extensive duration provides a significant advantage for in-depth studies into the evolution and dynamics of visual trends, artistic styles, and thematic content. By covering a broader range of temporal variations, our dataset allows for a more detailed analysis of how generative models respond to changing cultural, or seasonal influences over time. For instance, our preliminary investigation into the period around Halloween, specifically the seven days leading up to October 31, identified keywords like `Halloween`, `scary`, `costume`, and `pumpkin` much more frequently compared to other weeks. We plan to enrich the camera-ready version of our discussion by exploring deeper into the temporal aspects of our dataset, particularly how seasonal changes influence the content generated during different times of the year.
>
> > R1-Q2: The K-Means++ algorithm is employed to cluster the data into 1,000 groups, but it is not clear how the number 1,000 was determined. Furthermore, the impact of different K settings on the characteristics of the dataset should be subject to further analysis.
>
> We apologize for the typo; our dataset actually consists of 10000 clusters, as can be seen by the `cluster_id` column in the full dataset. To determine the optimal number of clusters, we employed the silhouette score method across various cluster sizes: {50, 100, 500, 1000, 2000, 5000, 10000, 20000}. The silhouette score measures how similar an object is to its own cluster (cohesion) compared to other clusters (separation). This experiment underscored that a configuration of 10000 clusters maximizes the silhouette score, indicating optimal internal similarity and external dissimilarity among the clusters (as can be seen from the table, we also tried 20000 clusters but it did not offer a significant deviation in silhouette score compared to 10000 clusters). This number of clusters provides the most meaningful and distinct categorization of our data, which is then used for KMeans++ clustering.
>
> **Silhouette Scores for Different Numbers of Clusters**
>
> | Number of Clusters | Silhouette Score |
> |-------------------|------------------|
> | 50                | 0.032            |
> | 100               | 0.043            |
> | 500               | 0.054            |
> | 1000              | 0.064            |
> | 2000              | 0.078            |
> | 5000              | 0.087            |
> | 10000             | 0.110            |
> | 20000             | 0.111            |

---

> > ### Comment · Reviewer_WApe · 2024-08-26
> >
> > Thanks for the detailed rebuttal from the authors.
> > I will increase the score.

---

> > > ### Author Rebuttal · Authors · 2024-08-26
> > >
> > > Thank you for your comment. We will update our camera-ready version to incorporate your feedback.

---

### Author Response · Authors · 2024-08-25
**Request for Further Feedback on our Rebuttal**

Dear reviewers, thank you for your insightful feedback. We are in the process of incorporating your suggestions and comments into our final manuscript. If you have any further questions or need additional clarification regarding our rebuttal responses, please do not hesitate to let us know. We would be happy to provide further details.

---

### Decision · Program_Chairs · 2024-09-26

**Decision:**

Accept (Poster)

**Comment:**

This paper introduces an extensive dataset featuring millions of user-generated images and styles, and presents a novel platform called Style Atlas, which leverages LoRA for personalized content generation and recommendations.

Reviewers praised the dataset’s comprehensiveness and its potential for studying visual generation trends, particularly in the current era of AIGC. They also raised questions about the dataset’s construction details, comparisons with existing datasets, and the analysis methods. The authors provided thorough responses to these inquiries. Ultimately, all reviewers agreed to accept the submission.